# Transduction Efficiency of Zika Virus E Protein Pseudotyped HIV-1*gfp* and Its Oncolytic Activity Tested in Primary Glioblastoma Cell Cultures

**DOI:** 10.3390/cancers16040814

**Published:** 2024-02-17

**Authors:** Jan Patrick Formanski, Hai Dang Ngo, Vivien Grunwald, Celine Pöhlking, Jana Sue Jonas, Dominik Wohlers, Birco Schwalbe, Michael Schreiber

**Affiliations:** 1Department of Virology, LG Schreiber, Bernhard Nocht Institute for Tropical Medicine, 20359 Hamburg, Germanyhai.dang.ngo@studium.uni-hamburg.de (H.D.N.); vivien.grunwald@haw-hamburg.de (V.G.); celine.poehlking@haw-hamburg.de (C.P.); janasue.jonas@haw-hamburg.de (J.S.J.); domink.wohlers@haw-hamburg.de (D.W.); 2Department of Neurosurgery, Asklepios Klinik Nord, Standort Heidberg, 22417 Hamburg, Germany; b.schwalbe@asklepios.com

**Keywords:** human cerebrospinal fluid, glioblastoma cell culture, lentiviral vector, retroviral vector, Zika virus, flavivirus, pseudotypes, transmembrane domain, gp41

## Abstract

**Simple Summary:**

Cells from the malignant brain tumor, glioblastoma multiforme (GBM) are highly heterogeneous. After tumor removal, some tumor cells remain at the tumor-brain boundary since in the brain, surgery cannot be performed with the normally required safety margin. Thus, it is of upmost importance to develop tools able to destroy remaining tumor cells. One strategy is to develop lentiviral vectors (LVs) with high specificity for GBM cells to transfer therapeutic genes into these cells. The Zika virus (ZIKV) provides an envelope with the protein E, which has a high specificity for GBM cells, making it a prime candidate for the development of LVs; so-called ZIKV protein E coated lentiviral particles. The study demonstrates that such LVs have an efficiency and high specificity for GBM tumor cells, leaving healthy cells mostly unharmed. These LVs open up new perspectives and therapeutic options for combating tumor cells that cannot be removed through surgery.

**Abstract:**

The development of new tools against glioblastoma multiforme (GBM), the most aggressive and common cancer originating in the brain, remains of utmost importance. Lentiviral vectors (LVs) are among the tools of future concepts, and pseudotyping offers the possibility of tailoring LVs to efficiently transduce and inactivate GBM tumor cells. Zika virus (ZIKV) has a specificity for GBM cells, leaving healthy brain cells unharmed, which makes it a prime candidate for the development of LVs with a ZIKV coat. Here, primary GBM cell cultures were transduced with different LVs encased with ZIKV envelope variants. LVs were generated by using the pNL*gfp*AM plasmid, which produces the lentiviral, HIV-1-based, core particle with GFP (green fluorescent protein) as a reporter (HIV*gfp*). Using five different GBM primary cell cultures and three laboratory-adapted GBM cell lines, we showed that ZIKV/HIV*gfp* achieved a 4–6 times higher transduction efficiency compared to the commonly used VSV/HIV*gfp*. Transduced GBM cell cultures were monitored over a period of 9 days to identify GFP+ cells to study the oncolytic effect due to ZIKV/HIV*gfp* entry. Tests of GBM tumor specificity by transduction of GBM tumor and normal brain cells showed a high specificity for GBM cells.

## 1. Introduction

Glioblastoma, also known as glioblastoma multiforme (GBM), is a highly malignant and aggressive brain tumor that develops from glial cells [1]. These tumors are known for their rapid growth and their ability to infiltrate the surrounding tissue, which makes them difficult to treat. The current standard treatment for high-grade gliomas is surgical resection. Unfortunately, the diffuse and invasive phenotype of glioblastoma means that some tumor cells almost always remain after surgery and can continue to grow. For this reason, radiotherapy and chemotherapy are the next stages of treatment. This treatment modality was introduced in 2005 after a modest prolongation of overall survival and progression-free survival was demonstrated [2]. Glioblastoma patients typically lived for less than one year after diagnosis. With therapeutic treatment applied, the 5-year survival rate ranges from 22% to 6%, depending on the age of the patients [3]. Various ideas are being pursued to combat glioblastoma. This includes the development of new cytostatic drugs, immune therapies and, above all, strategies that rely on tumor-directed killing processes, such as tumor-specific programming of natural killer cells (chimeric antigen receptor T-cell therapy) and the development of oncolytic viruses [4,5].

For oncolytic virus therapy, various viruses, including herpes simplex virus, adenovirus, measles virus, reovirus, and vaccinia virus, were modified in such a way that they can have an anti-cancer effect. Each of these viruses has its own characteristics in terms of tropism, route of infection, and replication in cells. In general, research in the field of oncolytic viruses is developing rapidly and there are a number of clinical trials investigating their potential against GBM [6]. The search for a therapeutic virus is driven primarily by the catastrophic prospects that still apply to a GBM diagnosis. Numerous studies have been carried out to date and some oncolytic studies have shown efficacy in vitro and sometimes partial efficacy in vivo in GBM-bearing mice. However, clinical studies promise some hope, but have not yet shown any explicit success [7].

Another approach based on the natural features of viruses is the concept of oncolytic pseudotypes targeting GBM [8]. Pseudotypes are modified viral particles in which the original envelope is replaced by an envelope from another virus [9]. The pseudotyping approach can be used to generate various changes, including a change in tropism towards the host cells. A pseudotyped particle based on human immunodeficiency virus type-1 (HIV-1) is also referred to as a lentiviral vector (LV). LVs are widely used in research and have shown promise for gene therapy applications as they are able to efficiently introduce genetic material into the DNA of both dividing and non-dividing cells [10].

Unfortunately, there are only a limited number of publications describing the development of GBM-specific HIV-1-based LVs. The development of LVs with the lymphocytic choriomeningitis virus (LCMV) envelope glycoprotein (GP) is particularly noteworthy [11]. It has been shown that LCMV-GP pseudotyped LVs exhibit increased infectivity to tumor cells in vitro, while other healthy primary human and brain cells of rats are not significantly affected [12]. However, pseudotyping of LVs with envelope proteins from viruses other than LCMV, like human foamy virus, rabies, Mokola or amphotropic murine leukemia virus has resulted in lower transduction efficiencies [13]. Thus, LCMV-GP should be seen as a promising candidate for LV-based cancer therapies [14]. In addition to LCMV-GP, the Japanese encephalitis virus (JEV)-prME envelope proteins were successfully used for pseudotyping of a gamma retrovirus [15] to study JEV neutralizing antibody [16].

To be successful, virus-based vectors against GBM must have a specific but also efficient entry mechanism while leaving non-cancerous brain cells unharmed. In this context, the Zika virus (ZIKV) with its special characteristics plays an important role [17]. It was found that babies born to mothers who had been infected with ZIKV were born with microcephaly. Microcephaly is characterized by an underdeveloped brain, which manifests itself in the form of an abnormally small head. The risk of microcephaly appears to be highest if a pregnant woman is infected with ZIKV in the first three months of pregnancy. A period in which the fetal brain cells multiply rapidly and in which various types of neurons and supporting glial cells develop from neural stem cells. At that stage, the fetal brain will be significantly impaired by the ability of ZIKV to infect and replicate in neural progenitor cells [18]. It is worth noting that the mother’s brain is not damaged and no signs of brain disorders have been detected related to ZIKV infection [19,20].

For this reason, ZIKV has a specific tropism for brain cells that divide rapidly, and these cells seem to express receptors that are weakly expressed or not at all present on differentiated and healthy brain cells, which divide only rarely or no longer at all [21]. There is an ongoing scientific debate about the relevance of certain cellular markers that act as ZIKV receptors in healthy non-tumor cells [22]. With regard to GBM, however, it is important that the two potential ZIKV receptor candidates, Axl/Gas6 and integrin αvβ5, can be found regularly on GBM tumor cells [23,24]. This makes GBM cells particularly permissive to ZIKV and thus, a prime target for oncolytic strategies [25].

For the development of ZIKV E-coated LVs, it is also important to investigate their infectivity in a cell culture system that comes as close as possible to the heterogenous in vivo situation. After removal of the tumor, some tumor cells remain at the tumor–brain boundary, as surgery in the brain cannot be performed with the usual safety margin. An LV should be able to target all of these different GBM cells.

In the past, it has been shown that cell lines grown in the laboratory are generally no longer representative of how tumors actually behave [26]. Therefore, primary cell cultures that reflect the heterogeneity of the different tumor cells as well as possible and are used as early as possible are well suited for the evaluation of LVs [27]. The addition of human cerebrospinal fluid (CSF) as a 50% supplement to the culture medium also supports the growth of a heterogeneous cell population by providing a medium that is as similar as possible to the original in vivo environment of the tumor [27,28,29]. Here we report on the transduction of primary GBM cells with different ZIKV E protein coated LVs and their toxic or non-toxic effects on tumor or non-tumor brain cells, respectively.

## 2. Materials and Methods

### 2.1. Plasmids and Cell Lines

Expression of ZIKV E proteins was carried out using a modified version of pcDNA3.1 (pME) as described earlier [8,27,30] and the pSVATGrev plasmid [31]. Green fluorescent protein (GFP) was expressed by plasmid pNL*gfp*AM (A. Trkola and N. Friedrich, Institute for Medical Virology, University of Zurich, Switzerland) [32]. pCMV-VSV-G (Addgene, #12260, Teddington, UK) was used for VSV-G expression. The following plasmids were used; pME-Z1 for expression of ZIKV prME envelope proteins [8], pME-ME for expression of ME (Δpr mutant) [27], pME-E2 for expression of E2 [30], and pE41.2 for expression of E^ΔTM^gp41^TMCY^ [30]. COS-1 cells (CVCL_0223) were provided by Friedrich Löffler Institute (Riems Greifswald, Germany). Human glioma cell lines U-87MG, U-138MG, and U-343MG were obtained from CLS Cell Lines Service (Eppelheim, Germany). All U-cells were cultured in DF medium (DMEM/10% fetal bovine serum, PAN-Biotech, Aidenbach, Germany).

### 2.2. Primary GBM Cultures

The study design was approved by the Ethical Commission of the Hamburg Medical Chamber (Ethik-Kommission der Ärztekammer Hamburg, Germany), registration number PV6041. Primary GBM cultures were established as previously described [8,27,30]. Non-cancerous brain tissue (AKH-22) was removed supratentorially from the border zone between gray and white matter during temporoparietal metastasectomy of colon carcinoma. Tissue samples were transported directly after surgical removal from the hospital to the cell culture unit at the Bernhard Nocht Institute for immediate processing. They were pressed through a 70 µm mesh (Fisherbrand, Schwerte, Germany) and the cell mixture was washed with 50 mL of DF medium. The cells were transferred into the respective medium and incubated in T75 cell culture flasks (Cell+™, Sarstedt, Nümbrecht, Germany) each containing 30 mL of (i) CSF, (ii) CSF-DF, or (iii) DF medium. Transmitted light images were taken using an M7000 microscope (EVOS, Thermo Fisher Scientific, Schwerte, Germany).

### 2.3. Immunostaining

Immunostaining of primary GBM cell cultures was carried out as described before [30]. In brief, cells were plated on 96-well microplates and fixed using 3.7% formaldehyde, then permeabilized by a treatment with 0.1% Triton X-100 in PBS buffer. Blocking was performed with PBS/0.5% Tween 20/5% BSA for 1 h at RT. Marker-specific primary mouse monoclonal antibodies were detected by using a secondary anti-mouse IgG H&L antibody in blocking buffer. Actin and cell nuclei were visualized using Phalloidin (iFluor 555 conjugate, AAT Bioquest, Biomol, Hamburg, Germany) and DAPI (Rotimount, Carl Roth, Karlsruhe, Germany). Microscopy was carried out using an EVOS FL Auto or M7000 Imaging system (Thermo Fisher Scientific, Braunschweig, Germany). Primary antibodies to Integrin, Mouse anti-integrin αvβ5 (P1F6) (Abcam, Berlin, Germany, ab177004); Axl, Mouse anti-Axl (C4-A8) (Thermo Fisher Scientific, Braunschweig, Germany, MA5-32897); Sox2, Mouse anti-Sox2 (9-9-3) (Abcam, Berlin, Germany, ab79351); Oct4, Mouse anti-Oct4 (GT486) (Abcam, Berlin, Germany, ab184665); Nanog, Mouse anti-Nanog (23D2-3C6) (Abcam, Berlin, Germany, ab173368); and Nestin, Mouse anti-Nestin (2C1.3A11) (Abcam, Berlin, Germany, ab18102) were used. The secondary antibody was Goat anti-mouse IgG H&L (Alexa Fluor^®^ 488) (Abcam, Berlin, Germany, ab150117). Actin was stained using Phalloidin-iFluorTM 555 Conjugate (Biomol, Hamburg, Germany, ABD-23153) and the nucleic stain was ROTI^®^Mount FluorCare DAPI (Carl Roth, Karlsruhe, Germany).

### 2.4. Production of ZIKV Envelope Pseudotyped LVs

Transfection of cells was carried out as described before [30]. Since the procedure for preparing ZIKV/HIV pseudotypes seems to be the most critical step of the study, we have specifically made an open access publication to provide easy access to our protocol explaining in great detail how we proceed in the preparation of pseudotype particles with a ZIKV-E coating [33]. It should be noted that this protocol is particularly designed to produce pseudotypes with the described plasmids. For pseudotype production, we are using COS-1 cells and two plasmids, each adjusted to a 1 mg/mL concentration. One plasmid for the expression of ZIKV envelope (plasmid pME) and the second plasmid for the expression of the HIV-1 core and its genome with green fluorescent protein (GFP) (plasmid pNL*gfp*AM) as a reporter for LV entry. ScreenFectA (Screenfect GmbH, Eggenstein-Leopoldshafen, Germany) was used as the transfection reagent in all experiments. The COS-1 cell culture supernatant containing the LVs was harvested 72 h post transfection by centrifugation (15,000 rpm, 5 min, RT, Eppendorf 5414, Hamburg, Germany) to remove cell debris and was used directly without any further modification or stored at 8 °C for up to 7 days.

### 2.5. HIV-1 p24-Antigen Assay

For p24 detection in LV-containing cell culture supernatants of COS-1 transfected cells, an in house p24-ELISA was used. In brief, ELISA plates (Maxisorb 96-well, Nunc, Thermo Scientific, Vantaa, Finland) were coated with 100 µL/well of anti-HIV-1-p24 D7320 (10 µg/mL; Aalto Bio Reagents, Dublin, Ireland). After blocking with PBSTM (PBS; 0.05% Tween 20; 3% low-fat milk), the plates were washed with PBST and incubated with 100 µL of the LV test solution for 1 h at room temperature (RT). The LV test solution was made by treatment of the LV COS-1 cell culture supernatant with Empigen (final conc. 0.7%, #30326, Merck, Darmstadt, Germany) and heat-inactivated at 56 °C for 30 min. After incubation of the LV test solution, plates were washed three times with PBST and bound p24 was detected using a goat anti-p24 polyclonal antibody horseradish peroxidase (HRP) conjugate (PA1-73094, Thermo Fisher Scientific, Dreieich, Germany) diluted 1:2000 in PBSTM (1 h, RT). Plates were washed three times with PBST and three times with PBS. For staining, to each of the wells, 50 µL of TBM (3,3′,5,5′-tetramethylbenzidine) substrate solution were added (SureBlue TBM substrate, Medac, Germany). After incubation for 15 min, 50 µL of a stop solution (1N HCl) was added. The density of the yellow staining was measured at 450 nm (Microplate reader MRXII, Dynex Technologies, Chantilly, VA, USA). The LV yield in COS-1 cell culture supernatants was in the range of 1.2–0.8 ng p24/mL for ZIKV/HIV*gfp* and 0.8–1.0 ng p24/mL for G-HIV*gfp* pseudotypes. Equal amounts of p24 equivalents were used in comparative transduction experiments.

### 2.6. Transduction of Tumor Cells by Lentviral Vectors

Two days prior to infection, tumor cells were seeded in 96-well plates (Cell+™, Sarstedt, Nümbrecht, Germany) in a total volume of 200 µL in CSF-DF medium to reach about 70% confluence on the day of infection. Since the primary GBM cultures have different growth characteristics, we seeded different dilutions of the cell suspension in 96-wells so that the best wells (70–80% confluence, 3500–4500 cells/well) could be selected for the transduction experiments on the second day. On the day of transduction, the medium was discarded and between 100 and 150 µL of LV-containing supernatant were added per well. Cells were incubated for 24 h at 37 °C and 5% CO_2_ and then media was exchanged with fresh CSF-DF medium. GFP+ events were monitored in daily intervals by an automated multichannel fluorescence life cell imaging system (M7000, Life Technologies, Fisher Scientific, Schwerte, Germany). For antibody-dependent neutralization, LV-containing supernatant was mixed with a dilution of ZIKV-positive human serum (provided by S. Kann [34], tested for ZIKV neutralizing hAb by a ZIKV plaque reduction assay) or mAb (rabbit anti-flavivirus group antigen antibody, clone D1-4G2-4-15, Novus, Bio-Techne GmbH, Wiesbaden, Germany) and added to the tumor cell containing wells. The amount of input LV was tested in the presence of 100 µg/well of a humanized monoclonal anti-PD1 antibody, used as a blocking reagent to prevent unspecific mAb effects in the test wells. For counting GFP+ positive cells, cell layers were stained using DAPI three days after infection and the wells were photographed at 4× magnification using the automatic scan function of the M7000 microscope (Thermo Fisher Scientific).

## 3. Results

### 3.1. Characterization of Primary GBM Cell Cultures

GBM tissue samples were transferred to cell culture in less than three hours after surgical removal of the tumor. We did not treat the tumor tissue with enzymes, we simply pressed it through a fine 70 µm sieve. After washing the cell suspension in medium, the cells were incubated in Cell+ culture flasks and monitored daily for the appearance of adherent cells. Three different media were used, standard DMEM/10% FBS (DF) pure human CSF and a 1:1 mix of DMEM/10% FBS and CSF (CSF-DF). The growth of the cells was monitored daily. The first cell passage was only carried out when the cells had reached a density of at least 50%. Growth rates of the various cell cultures are given in Table 1.

Since we experienced that cell growth was optimal in CSF-DF, we did not perform any growth in DF or CSF media from tissue probe 17 onwards. A time series of cell growth is shown in Figure 1 from a tumor tissue obtained from GBM patient 19, designated primary cell culture AHK-19.

From day one, cells of the adherent phenotype developed, which can be seen in the form of thin black lines in Figure 1a,b. Over time, a heterogeneous population of tumor cells develops with a variety of appearances, including cells with a characteristic roundish, ependymal, or astrocyte-like shape. In case of AKH-19, cells were used from day 14 (Figure 1f–i) for transduction experiments and characterization of molecular markers. It is important to note that every tumor cell isolation shows different growth rates, different forms of cell interactions and most importantly different fine structures and shapes, which illustrates the tumor term “multiforme” quite well. All AKH cultures as listed in Table 1 were positive for ZIKV receptors Axl and integrin αvβ5. Additionally, we also tested AKH cell cultures positive for Nanog, Nestin, Oct4, and Sox2 as shown exemplarily in Figure 2.

### 3.2. Transduction of Primary GBM Cells with Four Different LVs

In Figure 3a–d the transduction of primary AKH-16 cultures is shown for LVs carrying ZIKV prME, ME, E2, or E41.2 envelope proteins, respectively (details for the constructs are given in the Appendix A, Figure A1). Shown are full 96-well scans of transduced primary AKH-16 cultures and a corresponding section at 20× magnification.

In Figure 3a, Z1-HIV*gfp* transduced cells are shown. In the magnified section many rudimentary filopodial attachments are still visible at the cellular base. In contrast to its original form, however, the cell appears completely deformed, but has not yet collapsed completely as can be seen in other parts of the picture (white arrow). The large GFP+ cell is partially surrounded by tiny green dots, which might originate from filopodia degradation. The degradation of filopodia can be seen in more detail in Figure 3b,d. We also observed small GFP+ cells (white arrow). Since, on day three, these small GFP+ cells have a nucleus, they cannot originate from filopodia fragments. Comparing this appearance with the immunostaining images of AKH-16 (Figure 2f,g), one can clearly see the altered cellular shapes in the LV treated AKH-16 culture. Figure 3b shows a 96-well scan of an ME-HIV*gfp* LV treated culture. The deformation and collapse of the filopodia-positive cells is obvious in this example and clearly visible in the magnified section. The cells appear to retract their filopodia and a condensed large sphere is formed (white arrows). The filopodia collapse into much smaller globules, and the entire structure looks like little beads on a string. This is a typical image showing the destruction of the highly complex fine structure of astrocyte-like cells. Since the green spheres (white arrows) still have a nucleus, they cannot originate from filopodia fragments which are visible as tiny green dots. In Figure 3c, AKH-16 cultures are transduced using E2-HIV*gfp*. This example shows two deformed cells, but also one cell that has barely retained its structure (white arrow). The green circular GFP+ cell in the upper left corner of the image (white arrow) shows the typical shape of one of the actin-positive AKH-16 cells. These actin+ cells are usually flat and spread out, but in contrast to their original shape, they look like tiny fried eggs after successful transduction. Although this cell appears to be rather intact, the corresponding GFP+ signal extinguished after two weeks, and the cell finally died as did all other cells that were GFP+. In Figure 3d, cells are transduced by E41.2-HIV*gfp* showing the highest rate of transduction compared to Figure 3a–c. The enlargement shows an example of a disruption of the cellular network, which is normally observed very well in AKH-16 cultures. The fragmentation of the remaining filopodia, leading to the formation of tiny globules is also very well seen here. The normally adherent, planar spreading of the healthy cells is no longer present and the collapse into a smaller sphere becomes visible. A GFP+ cell showing the adherent phenotype is seen in the lower left corner (white arrow). This cell also collapses and its GFP+ signal disappeared after 13 days, as does the GFP signal of all other cells. Overall, transduction with HIV*gfp* derived LVs and transfer of the LV genome expressing GFP under the control of the CMV promoter leads to destruction of cell integrity and eventual death of GFP+ cells. The infection study with four different ZIKV envelopes also showed that E41.2-HIV*gfp* is the most efficient LV compared to the three LVs with envelopes which have the original ZIKV TM (E^TM^) domain. Replacement of E^TM^ with the HIV-1 gp41-TM and gp41-CY domains appears to be advantageous for the efficiency of the E41.2-HIV*gfp* pseudotype construct.

### 3.3. Neutralization of LV Entry into GBM Tumor Cells

To verify ZIKV E-specific LV entry, we used a Flavivirus group-specific antibody (4G2) and sera positive for ZIKV neutralizing antibodies. Neutralization was achieved by simultaneous addition of LV and antibody to cells, which were then exposed to the antibody/LV mixture for 24 h. By this procedure, LV neutralization was tested in quadruple 96-wells. Neutralization was determined on the third day by counting GFP+ cells in each of the four wells. A 90% reduction titer was observed in a range of 80 to 400 for the two human ZIKV antibody positive sera and between 2000 and 3000 for the 4G2 monoclonal antibody. When compared, ME-HIV*gfp* was more sensitive to antibody-dependent neutralization than the E41.2-HIV*gfp* pseudotype. In Figure 4a, a complete scan of an ME-HIV*gfp* infected AKH-11 culture is shown. Pseudotype transduction was carried out in the presence of 100 µg of an anti-PD1 antibody. This determined the initial quantity of GFP+ LV entry events (GFP+ foci). Figure 4b showed exemplarily the initial quantity for the respective test and in Figure 4c one of the four wells that were tested for each of the three sera. The number of GFP+ cells was counted in all four wells and their mean values are shown in Figure 4d. Neutralization of E41.2-HIV*gfp* is shown in Figure 4e–g demonstrating again the E-specific blockade of pseudotype entry.

### 3.4. Efficiency of LV Entry in Primary GBM Cell Cultures

Five different AKH cell cultures were transduced with E2- or E41.2-HIV*gfp* LVs. On day three transduced cells were washed and stained using DAPI. Blue-stained nuclei and green fluorescent cells were counted to determine the percentage of infected cells. Infection rates were compared to G-HIV*gfp* infected cells. A section of an AKH-16 cell culture stained blue and green is shown as an example in the Appendix A, Figure A2. Complete scans of 96-wells for E41.2-HIV*gfp* and G-HIV*gfp*, ZIKV E versus VSV-G, are shown in Figure 5.

As shown exemplarily in Figure 5, 830 GFP+ cells were counted for E41.2-HIV*gfp* and 150 for G-HIV*gfp*. For each measurement, four wells were counted out to determine the efficiency of the LVs in terms of their infectivity. The means and standard deviations for E2-, E41.2-, and G-HIV*gfp* efficiencies are shown in Table 2.

For comparison, E2- and E41.2-HIV*gfp* were chosen, since they are identical in their E and stem part and only differ in the TM region. As shown in Table 2, the best transduction results were observed for E41.2-HIV*gfp*, showing rates of GFP+ cells 2–6× higher compared to G-HIV*gfp*. Especially, the AKH-16 cell culture showed cells of higher permissiveness. Compared to G-HIV*gfp* the mean GFP+ number of AKH-16 cells was 4× higher. In these experiments E2-HIV*gfp* showed similar GFP+ rates as with G-HIV*gfp*.

The only exception being AKH-13. Of all five AKH cultures, AKH-13 had a GFP+ rate of approximately 3%, which was relatively low compared to the other AKH cultures. In contrast, however, the differences to the G-HIV*gfp* control were highest for AKH-13, both for E2-HIV*gfp* and E41.2-HIV*gfp*, with a 7- and 6- fold increase in GFP+ cells, respectively. Due to the heterogenous nature of the primary cell cultures, the transduction experiments should show variations in the number of GFP+ cells. But in all transduction experiments, GFP+ rates were close or significantly higher as with the commonly used VSV-G pseudotyped LV, G-HIV*gfp*.

### 3.5. Oncolytic Activity of ZIKV/HIVgfp Lentiviral Vectors

Five 96-wells were transduced for each LV and the numbers of GFP+ cells were monitored over a period of 9 days. As can be seen in Figure 6, the rates for GFP+ cells decreased rapidly from the 5th day after transduction.

In agreement with the observation shown in Figure 2a–d, we also observed loss of cell integrity and the collapse of the fine cellular morphology of the GBM cells. GFP+ signals were peaking generally on day three and were vanishing more rapidly from day four. Thus, ME- and E41.2-HIV*gfp* transduction induced cell death, in contrast to brain cells of a non-cancer origin, designated AKH-22. GFP+ signals in AKH-22 cells was low with less than 6 GFP+ cells/96-well containing about 1000 adherent AKH-22 cells. Interestingly, G-HIV*gfp* caused more GFP+ cells, albeit in low numbers (11–30 GFP+ cells/well), but with a significantly higher number than the ME and E41.2 candidates. This demonstrates also the high specificity of ZIKV/HIV-pseudotypes for GBM cancer cells leaving healthy brain cells mostly unaffected whereas VSV-G-coated pseudotype entry into AKH-22 was more evident.

## 4. Discussion

### 4.1. GBM Primary Cell Cultures

Establishing primary cell cultures from GBM tissues is critical for conducting research and especially for the development of LVs aimed at understanding the disease and developing potential anti-GBM strategies [35]. Primary cultures provide a closer representation of the tumor cell’s characteristics compared to established laboratory-adapted cell lines or GBM cells that have been cultivated in vitro for a long time. In this study, all primary cell cultures of glioblastoma cells are derived directly from patient tumors and are given into in vitro cell culture conditions within <3 h after surgery. Once the cultures produced a sufficient amount of tumor cells, they were characterized for expression of ZIKV entry receptors Axl/Gas6 and Integrin αvβ5. Axl/Gas6 was detected regularly and evenly distributed on GBM cells. In contrast, the presence of integrin was preferentially targeted to the nucleus and was particularly pronounced in dividing cells [30]. In addition to the ZIKV receptors, we further characterized all cell lines for the expression of embryonic stem cell markers Nanog, Nestin, Oct4, and Sox2, as these are critical receptors for the progression of various human malignancies, including brain tumors [36,37,38,39,40]. Additionally, to the proposed ZIKV receptors, it has been shown that Sox2 is critical for ZIKV GBM infections [41].

The LV transduction tests were carried out in parallel with the first tests, which could generally be performed from the second week onwards. For LV development, we do not work with older, >6 months, AKH cultures. We therefore continuously collect tumor samples in order to create new GBM cultures for LV transduction studies. The approach is to use heterogeneous GBM cultures without transforming or selecting them, so that the cells do not deviate significantly from their original state. The addition of human cerebrospinal fluid as a major cell culture medium supplement is also part of this approach, which aims to bring the cell culture conditions as close as possible to the original in vivo situation [42]. The multiforme character of the tumor is also one of the reasons why we keep using new cultures. Especially because the tumor is very heterogeneous, it must be shown that an LV is also able to transfer therapeutic genes into the cells from various tumors. In our experience, primary GBM cell cultures show a very detailed morphology of the brain cells compared to older GBM cells or laboratory strains that have lost this appearance. Because in primary cultures, the isolated tumor cells form such excellent, fine structures, they are well-suited to observing LV entry and especially tumor cell destruction. However, the cell culture system is limited to the presence of adherent cells, and it would be interesting to study the entry of LVs also in 3-D cell culture models if, for example, neurospheres are less or more sensitive to LV entry compared to 2-D cultured GBM cells. In our opinion, 3-D models are better suited for the analysis of replication-competent oncolytic viruses or for the analysis of drugs and glioblastoma biology in general. LV-induced morphological changes or apoptotic effects of individual cells can preferably be analyzed in 2-D cell culture systems. Especially in GBM research, attention must be paid to the interactions to be investigated in order to select the appropriate cell culture system [43].

Comparing LV transduction in tumor and non-tumor cells, we observed a higher specificity of ZIKV/HIV*gfp* for tumor cells as compared with VSV/HIV*gfp*. These experiments are limited to the region from which the healthy brain cells were taken, indicating that healthy brain cells do not appear to be a major target for ZIKV/HIV*gfp*. However, the tests with healthy brain cells presented in this study are limited to a single tissue sample and are therefore, probably not sufficient to completely rule out possible effects on the heterogeneous cell populations in the brain. We therefore cannot exclude the possibility that a ZIKV/HIV*gfp* LV would infect neural stem cells or other healthy cell types in the brain, which could lead to possible off-target effects [44,45,46]. Given the poor 5-year survival rate in GBM, this may not be a major problem, but off-target effects should be avoided as much as possible. In general, off-target effects must be considered in a differentiated manner. On the one hand, the specificity of cell entry and on the other hand, the oncolytic mechanism of action must be taken into account. Currently, the LV specificity for GBM cells is given by the ZIKV E protein, and it is not clear what kind of changes to this protein could further improve the differentiation between tumor and normal cells [45]. The oncolytic mechanism of action is therefore another important factor that should preferably be directed against tumor cells. The GFP expression might be an example of how cells can be inhibited, it is therefore an interesting question if other genes can be identified that are accepted by normal cells, but lead to cell death when expressed in GBM cells. In addition, LVs can be used for the transfer of siRNA or mutation-specific sgRNA, which are preferably effective against GBM cells [47,48]. Therapeutic genes of this kind can easily be inserted into retro- or lentiviral genomes [49,50]. In combination with ZIKV E, a very high specificity can be achieved, which might greatly reduce any off-target effects by such LVs.

### 4.2. Virus Envelope for Pseudotyping Lentiviral Vectors

One of the major hurdles has been how to target a gene transfer vector to cells of interest. With the technique of pseudotyping, virus particles can be generated with viral envelope proteins from another virus to either restrict or broaden the host cell tropism. Pseudotyping is mainly used for the enveloped viruses of lentiviruses (HIV-1), retroviruses (murine leukemia virus) and rabies viruses (VSV). Lentiviruses, as a subgroup of retroviruses, provide the advantage of being able to transfer genetic material into both dividing and non-dividing cells, mainly due to the HIV-1 accessory protein *vpr* [51]. They also offer a much larger cargo capacity compared to other viral vectors, so that they can carry and transfer relatively large therapeutic genes or multiple genes simultaneously. This feature is an important advantage that significantly expands their range of applications [52].

A commonly used envelope protein for pseudotyping is the vesicular stomatitis virus G protein (VSV-G), which is well suited for infection of many cell types, as the envelope receptor pairings such as VSV-G—phosphatidylserine (PS) or VSV-G—low-density lipoprotein receptors (LDLR) offer a broad tropism. From a technical point of view, like G-HIV*luc* or G-HIV*gfp* as used in the present study, have the advantage that they are stable and can be enriched by ultracentrifugation to generate high titers [53]. This is one of the reasons why VSV-G is often used for ex vivo cell transduction and the industrial processing of VSV-G coated LVs. Since we cannot be sure and there is no data available on the stability of ZIKV/HIV pseudotypes, we are not modifying LV-containing cell culture supernatants. Adjusting LV extracts by counting GFP+ foci or by analyzing the HIV-1 p24 antigen of the LV extract seem to be the method of choice to normalize LV extracts.

It was noted that vesicular stomatitis virus G-protein (VSV-G)-pseudotyped lentiviruses which do not utilize a specific receptor-dependent entry pathway have a much wider transduction potential. VSV-G-pseudotyped HIV-1 was used to transduce patient-derived GBM laboratory-adapted cell lines with an apparent 100% success rate [6]. But from the reference, it is not clear how the VSV-G pseudotyped HIV-1 was generated or concentrated to reach such a high rate of transduction. We prefer to use unmodified LV-containing supernatants from the originally transfected cell cultures for comparing efficiencies between different LVs. It is evident that LVs produced, harvested, and finally concentrated by transfection from much larger culture vessels than 24-wells must result in a higher number of i.e., GFP+ cells when added to cells in a 96-well. The development of a more efficient LV packaging method is certainly one of the next tasks to be addressed. However, with regard to combatting GBM, the broad tropism of VSV-G is undesirable for planned in vivo applications due to the risk of severe off-target infectivity, as was also shown in our study.

To transfer therapeutic genes into GBM tumor cells, there is a special need for an envelope protein on the LV surface that has a high binding activity to a corresponding receptor present on the GBM tumor cell. Fortunately, the receptor molecules Axl/Gas6 and integrin αvβ5, proposed for Zika viruses, are particularly exposed on GBM tumor cells, which makes ZIKV envelopes perfect candidates for the encapsulation of retroviral vectors. For the development of an anti-GBM pseudotype, the search for a viral envelope with a perfect tropism, like ZIKV E, is only one part of the solution. A second aspect is the assembly of the pseudotype particle. The prerequisite for this is that both the selected envelope protein and the HIV-1 proteins gag and gag-pol localize at the cell membrane. However, HIV-1 is assembled at the outer cell membrane, whereas ZIKV is not. When using ZIKV E, it must therefore be modified so that it becomes part of the normal HIV-1 assembly process. In addition to the publications of our group, there is one publication that describes such a modification, in which the transmembrane region of ZIKV E was exchanged for that of VSV-G [54]. Together with a retroviral vector, they showed also an enhanced transduction rate for tumor cells in comparison to the use of VSV-G. Since we are using a lentiviral vector system based on HIV-1, we decided to use the transmembrane region of gp41 (TM^gp41^) and its cytoplasmic domain (CY^gp41^) as a replacement for TM^E^ [30]. Here, we have shown that by using such a construct, E^ΔTM^TM^gp41^CY^gp41^, a significantly better transduction of different primary GBM cultures could be achieved compared to VSV-G.

Efficient assembly and budding of the pseudotype particle at the cellular membrane are important aspects on the way to a functional ZIKV/HIV pseudotype. The budding of the virus particle is a process driven by the ESCRT machinery [55]. This requires the presence of a specific PTAP sequence motif within the cytoplasmic part of the envelope that links it to the ESCRT-I factor TSG101. Viruses like Ebola (matrix protein), lymphocytic choriomeningitis virus (LCMV, Z protein), Lassa (Z protein), Moloney murine leukemia virus (MMLV, p12-*gag*), simian immunodeficiency virus (SIV, p6-*gag*), human T-lymphotropic virus type 1 (HTLV-1, p19-*gag*) and HIV-1 (p6-*gag*), hijack all ESCRT-I via their PTAP sequence [56,57]. The TM or CY domains of HIV-1 gp41 have no such epitope, but the p6 domain of HIV-1 *gag* hijacks ESCRT-1 via its PTAP epitope. Thus, the TM and CY domains of HIV-1 seems to be well suited as a replacement for the E TM domain.

Another aspect is a fundamental problem with oncolytic virus strategies. A safe therapeutic virus that replicates in the human brain needs not only an efficient on-switch; it also needs a very effective off-switch. The concept of oncolytic pseudotypes offers a very effective off-switch, as it only leads to pseudotype entry, also known as single-round infection, and most importantly, all transferred genes are deleted together with the transduced cells whenever the oncolytic effect is strong in this LV.

### 4.3. GFP as a Reporter for Transduction

Green Fluorescent Protein (GFP) is commonly used as a selection marker for gene transduction and to track tumor or stem cells. In addition, GFP is not considered toxic to living cells in most cases. However, the potential for toxicity can depend on how GFP is used and expressed in cells [58]. Some cells have shown to be sensitive after being transfected by various GFP-plasmid vectors when GFP is expressed under the control of SV40 or CMV promotors. These include mouse embryonic and baby hamsters kidney fibroblasts, whose GFP fluorescence disappears within 4–5 days, and which also die after this period [59]. Initiation of apoptosis has been postulated as a possible mechanism for GFP-dependent toxicity and cellular death. The DAPI staining technique in this study shows also that nuclear decay was observed in HIV*gfp* pseudotype-infected cells. We observed roundish GFP-positive spheres under fluorescence microscopy, which did not correlate with concomitant positivity for DAPI as shown in Figure 7.

The GFP+ cells shown in Figure 7 are not observed after further days, so that after about 13 days, no GFP positivity could be observed anymore in the LV transduced GBM culture. Condensed black structures of dead cells are then present at these sites, clearly negative for GFP and DAPI. From our observations, we conclude that cells transduced with the HIV*gfp* pseudotype lose their structure, collapse, and eventually die. This effect was not only a rare event, but death of infected cells was observed in all primary cell cultures after a period of 7–10 days. This observation and its time course is similar to the GFP-induced cell death observed in mouse embryonic fibroblasts and in baby hamster renal fibroblasts seen by other researchers [59]. In the context of GFP-induced toxicity and apoptosis, various morphological changes such as loss of structural integrity were also observed [59].

In stem cell biology, stable expression of fluorescent reporter genes, such as GFP, is of great importance to follow the development of stem cells and their progeny. Interestingly, it has been demonstrated that cells with GFP as reporter are not suitable for studies on longer time scales, due to the cytotoxicity of GFP [60]. In primary cortical neuron cultures, transfer of the GFP reporter gene triggered apoptosis, demonstrating a high GFP-dependent toxic effect [61]. In addition, rat hepatic adult stem cells were shown to be highly sensitive to GFP-induced damages, preventing the establishment of GFP-expressing strains [60]. Studies in mice showed that GFP co-expressed together with ß-galactosidase, induced growth retardation of neurons and premature death due to the apoptotic effects observed in the forebrain area [62]. It is proposed that the apoptotic effects seen are in part due to the oxidative stress produced by GFP [63]. Additionally, in neuroblastoma cell lines, GFP expression has been shown to induce oxidative stress, resulting in selection of cells sensitized to death [64].

One of our original ideas was to isolate the infected cells in order to further characterize them. In agreement with the reports of [60], we were also unable to generate clonal lines from the pseudotype infected, GFP-positive tumor cells, which prevents use of techniques like FACS or Macs sorting. We have not yet identified a fluorescent marker that enables long-term cultivation of GBM tumor cells. In particular, the expression of mCherry proved to be more toxic compared to GFP, which is also consistent with observations from other studies [65]. Interestingly, there is a new report on a fluorescent protein that appears to be suitable for the long-term labeling of fine structures, like filopodia, in neurons [66]. Whether E2-crimson protein can be of help as a fluorescent marker, constitutively expressed in cells from primary GBM cell cultures, to establish classical cell lines is a question that still needs to be addressed.

Another aspect that is important in the context of the present study is the influence of GFP expression on the structural integrity of cells. Impairment of the actin-myosin interaction has also been reported due to GFP cytotoxicity. Transduction of myotubes that were derived from primary myoblasts with GFP lentivirus vector showed an impaired cell performance. Expression of GFP resulted in disruption of actin-myosin interaction and consequently impaired contractile function in transduced cells [67,68].

Considering the various GFP-induced defects, it is likely that GFP may be the trigger for the damages we commonly observe in GBM tumor cells after HIV*gfp* pseudotype transduction. For example, intact and healthy fine structures of astrocyte-like cells, as seen in the immunostaining images in Figure 2, are no longer detectable after HIV*gfp* pseudotype entry. In this context, it has been shown that GBM astrocytes, by targeting their immunometabolic pathway, cause destruction of the GBM tumor architecture [69].

In agreement with the studies described by Liu et al. [59], we also observed that after five days the GFP+ signal in the transduced cells began to vanish. The non-transduced cells, on the other hand, continued to multiply unhindered. The GFP effect on GBM cells, identified as a new side effect, can be interpreted as an oncolytic effect. Thus, the ZIKV/HIV*gfp* LVs would not only be a tool for the transfer of therapeutic genes into GBM cells but would also have a general oncolytic potential due to its GFP expression.

## 5. Conclusions

For oncolytic virus therapy, the biggest hurdle is to transfer the relevant genetic material into a sufficient number of target cells and to avoid the transduction of non-target cells. It is important to design the gene transfer vector in such a way that it has the best possible specificity for the desired cells. ZIKV envelope coated LVs provide prime tools for targeting GBM since they have a high specificity for these kind of tumor cells.

## Figures and Tables

**Figure 1 cancers-16-00814-f001:**
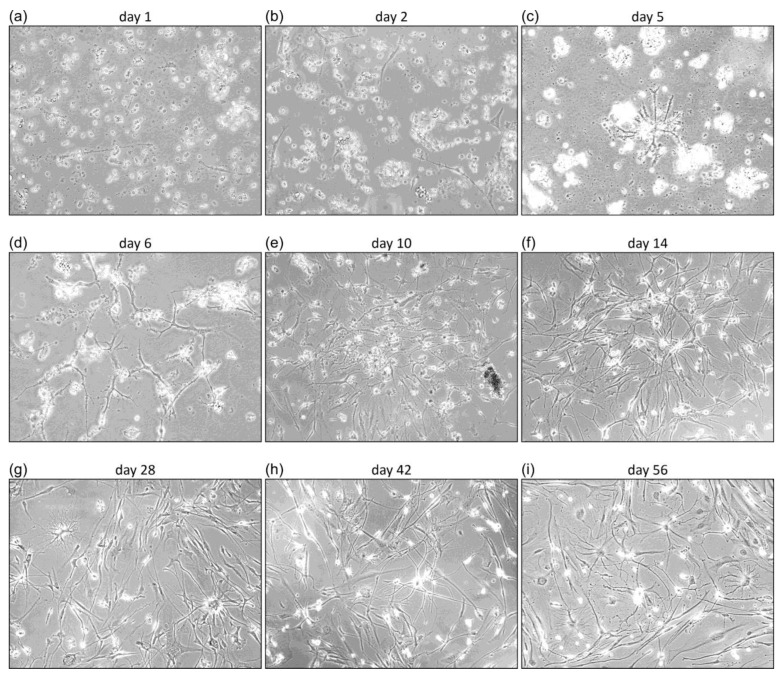
Growth of AKH-19 tumor cells in CSF-DF medium at different stages of development. Day 1, the cell suspension after processing the tissue. Days 14–56, cell cultures used for pseudotype infection experiments. (**a–i**) Cell cultures are shown at a 10× magnification (EVOS M7000 microscope).

**Figure 2 cancers-16-00814-f002:**
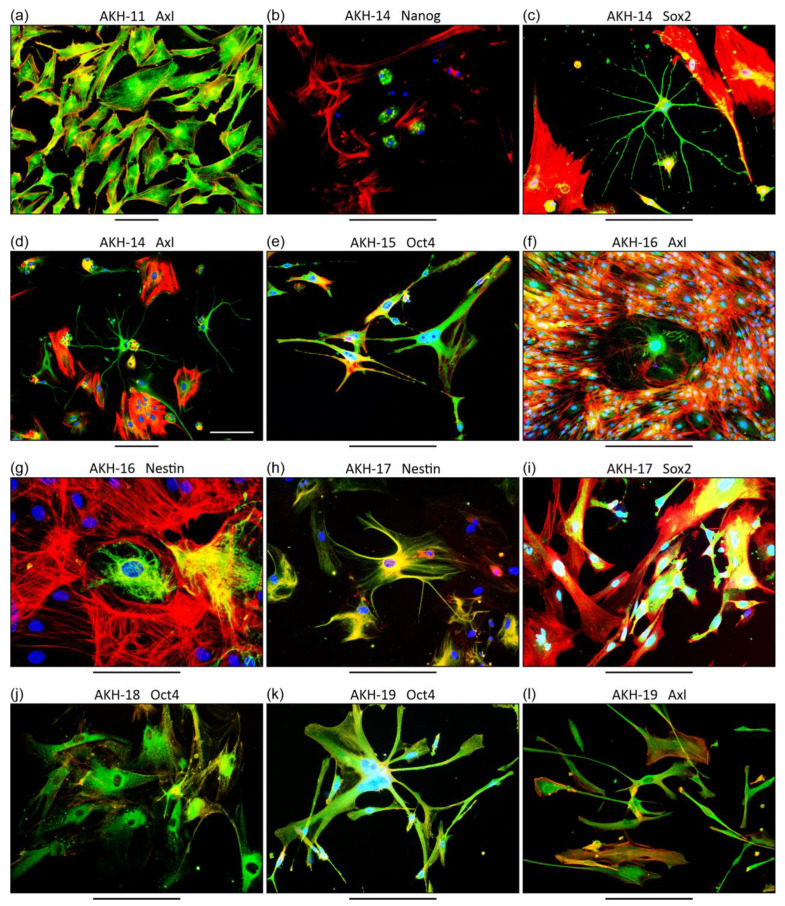
Immunofluorescent staining of representative markers in primary GBM cell cultures. (**a–l**) Examples of the different morphologies, and established cell-to-cell connections in primary GBM cultures, which all form their own appearances. Commercial marker-specific monoclonal antibodies were used for immunostaining and were detected by a goat anti-mouse IgG antibody (Alexa 488, green). Red, phalloidin actin staining. Blue, DAPI staining of nuclei. Scale bars = 200 µm. AKH-XX = primary tumor cell cultures from different GBM patients.

**Figure 3 cancers-16-00814-f003:**
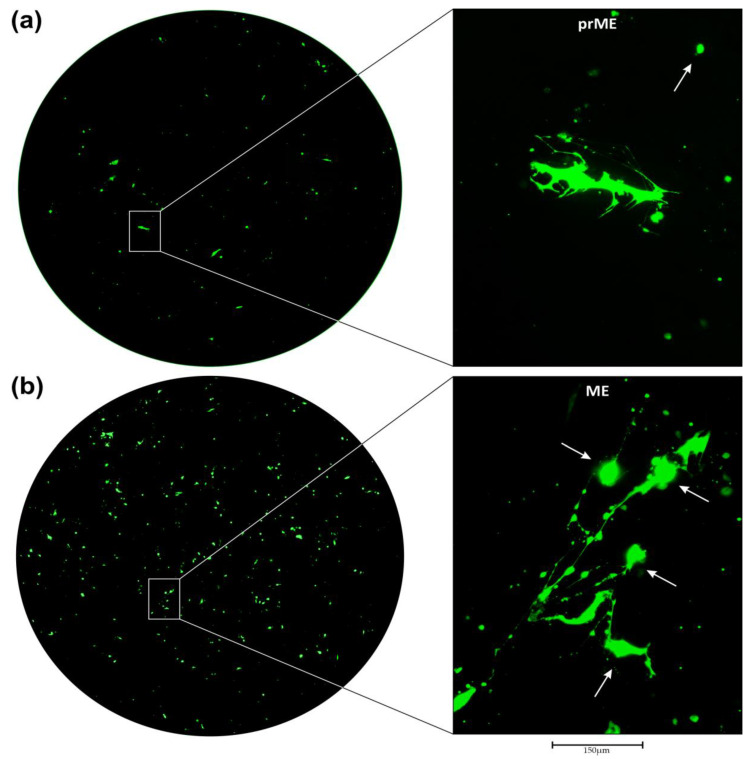
Infection of primary AKH-16 cultures with ZIKV/HIV*gfp* LVs. Green fluorescence is the indicator for a successful transduction. For transduction, tumor cells were grown in CSF-DF medium and incubated with cell culture supernatants from COS-1-transfected cells. **Left**: Complete overview of a pseudotype infected cell culture in a 96-well at 4× magnification (well Ø = 6.9 mm). **Right**: View of a section at 20× magnification. Transduction of cells with (**a**), Z1-HIV*gfp* (prME envelope), 115 GFP+ cells; (**b**), ME-HIV*gfp* (ME envelope), 420 GFP+ cells; (**c**), E2-HIV*gfp*, (E envelope), 350 GFP+ cells; (**d**), E41.2-HIV*gfp* (E^ΔTM^gp41^TMCY^ envelope), 525 GFP+ cells. GFP+ cells were detected on day 3 using an EVOS M7000 microscope with a 470/525 nm filter (excitation/emission).

**Figure 4 cancers-16-00814-f004:**
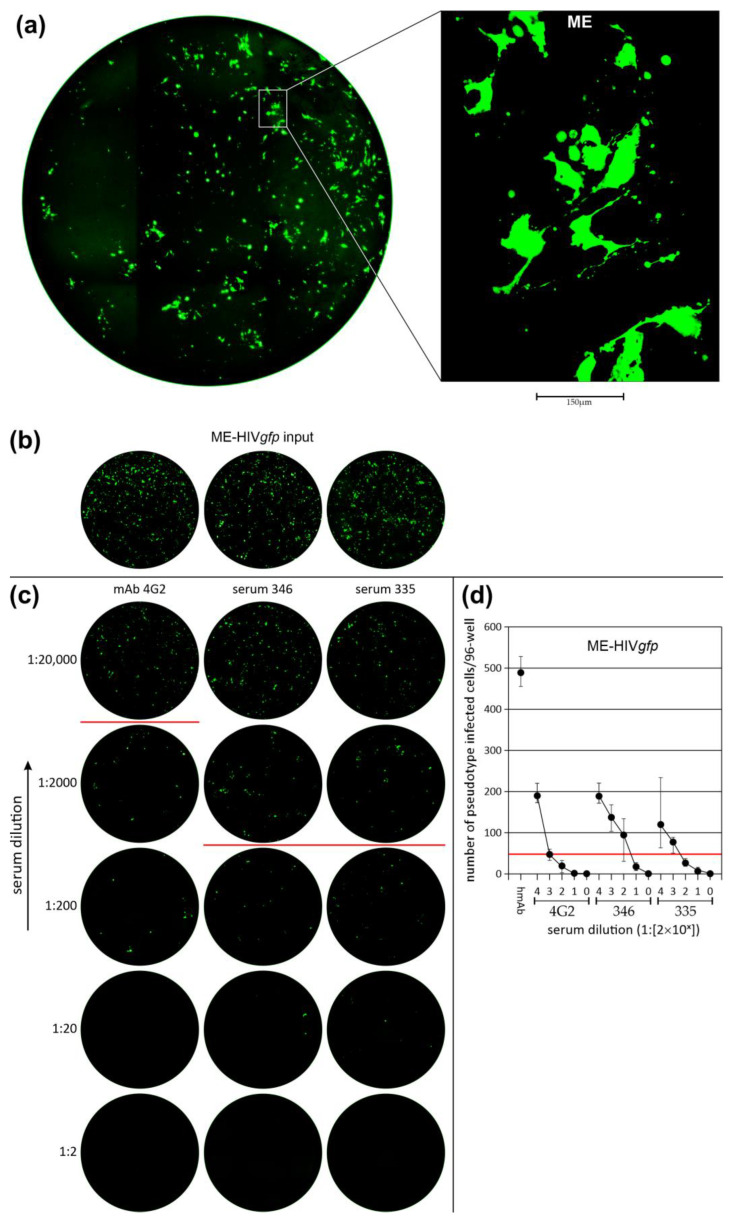
Neutralization of LV entry into GBM tumor cells. Each neutralization experiment was carried out in quadruple (4 well per antibody dilution). In the figure, for each neutralization test one of the four wells is shown as an example. (**a**), AKH-11 primary culture transduced with ME-HIV*gfp*. (**b**), ME-HIV*gfp* initial quantity of GFP+ cells in the presence of 100 µg PD1-hmAb/well with an average of 484 ± 35 GFP+ cells/well. (**c**), For each neutralization test, a complete scan of one of the four infected wells and one of the input wells is shown as an example. Neutralization was monitored by GFP expression, and the number of GFP+ cells were counted for each well. (**d**), the black dots represent the mean value of GFP+ cells from the 4 wells. The upper and lower bars show the highest and lowest number of GFP-positive cells identified. (**e**), initial quantity of E41.2-HIV*gfp* induced GFP+ cells with an average number of 715 ± 38 cells/well. Transduction was carried out in the presence of 100 µg/well hmAb as for ME-HIV*gfp*. (**f**), for each serum dilution, one of the four 96-wells is shown as an example. (**g**), black dots represent the mean of four neutralization assays. The bars indicate the highest and lowest number of GFP+ cells/well. (**d**,**g**), the red line indicates the level of 90% reduction of LV entry. PD1-hmAB = anti-PD1 humanized monoclonal antibody (2 mg/mL); 4G2 = anti-flavivirus group antigen antibody clone D1-4G2-4-15 (1 mg/mL); 346 and 335 = human ZIKV antibody positive sera (346, PRNT90 = 1/2048; 335, PRNT90 = 1/1024 determined using Vero-B4 cells).

**Figure 5 cancers-16-00814-f005:**
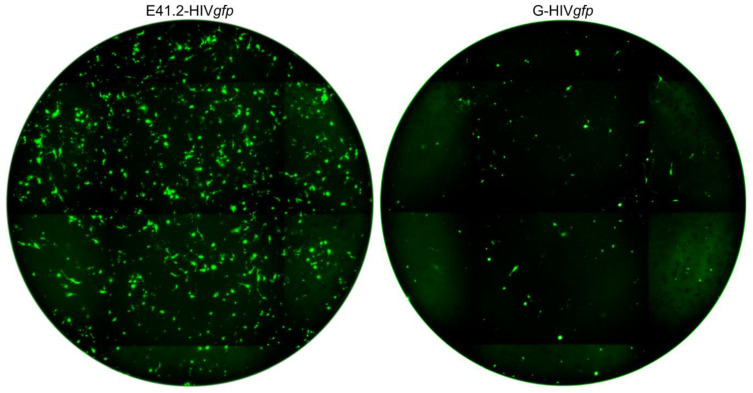
Efficient transduction of AKH-16 cells using E41.2-HIV*gfp* and G-HIV*gfp.* Shown is a complete overview of a 96-well at 4× magnification (well Ø = 6.9 mm) three days after LV transduction. Number of GFP+ cells: E41.2-HIV*gfp* = 830/well, G-HIV*gfp* = 150/well.

**Figure 6 cancers-16-00814-f006:**
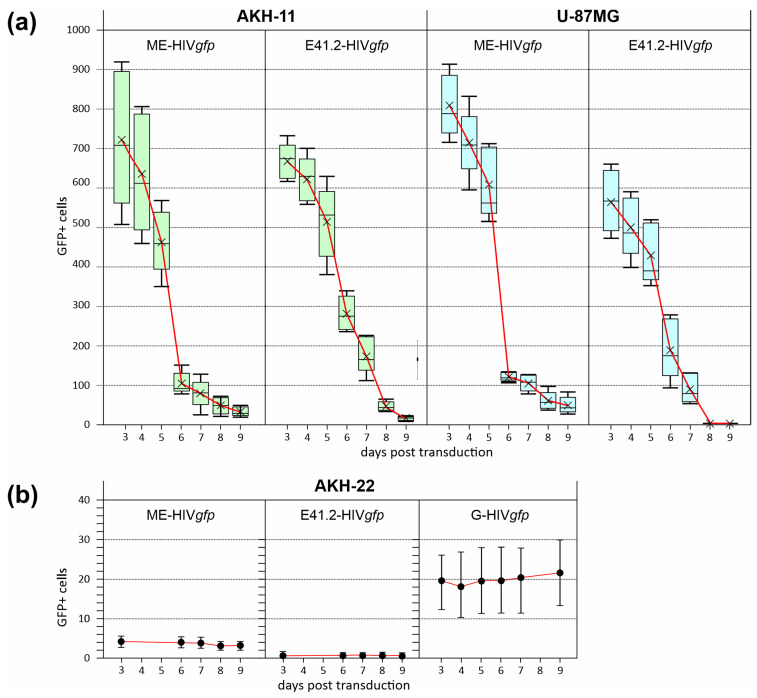
Monitoring of LV-transduced cell cultures over a period of 9 days. (**a**), ME- or E41.2-HIV*gfp* transductions of AKH-11 (green) and U-87MG (blue) tumor cells were carried out in five separate 96-well and the GFP+ cells were counted. (**b**), transduction of AKH-22, non-tumor primary brain cells (black dots). AKH-22 cells were cultured in CSF-DF medium and used for transduction two days after sampling.

**Figure 7 cancers-16-00814-f007:**
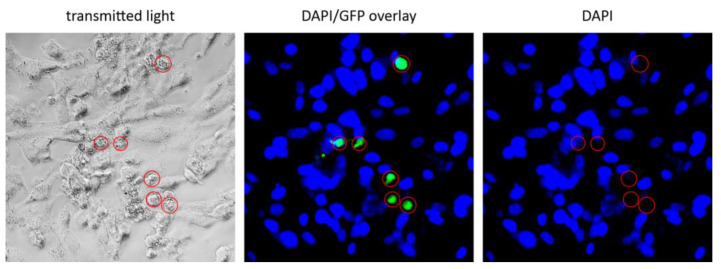
Negative DAPI stain of GFP+ cells after LV transduction. Shown is an example of GFP-positive cells displaying negative DAPI staining (day 6). Red circle, cells positive for GFP and negative for DAPI.

**Table 1 cancers-16-00814-t001:** Growth rates ^1^ of freshly isolated cells in different media ^2^.

AKH Cells	DF	CSF	CSF-DF
01	++	−	+++
02	++	+	++
05	++	+	+++
09	+++	+	+++
10	++	+	+++
11	++	−	+++
12	+++	+	+++
13	++	+	n.d.
14	++	+	+++
15	+	−	++
16	++	−	+++
17	++	−	+++
18	n.d.	n.d.	+++
19	n.d.	n.d.	+++

^1^ First cell passage: −, more than 5 weeks; +, 3–5 weeks; ++, 1–3 weeks; +++, <1 week. N.d., not done. ^2^ DF = DMEM/10% FBS; CSF = human cerebrospinal fluid; CSF-DF = 50% CSF/45% DMEM/5% FBS.

**Table 2 cancers-16-00814-t002:** Efficiency of E2-HIV*gfp* and E41.2-HIV*gfp* transduction compared to G-HIV*gfp*
^1^.

Cell Cultures	E2-HIV*gfp*	E41.2-HIV*gfp*	G-HIV*gfp*
AKH-13	3.6 ± 0.2	3.2 ± 0.3	0.5 ± 0.1
AKH-14	6.3 ± 1.1	11.7 ± 0.7	5.4 ± 0.5
AKH-16	8.2 ± 0.6	24.0 ± 1.3	5.9 ± 1.6
AKH-17	10.0 ± 3.7	n.d.	6.8 ± 1.1
AKH-18	5.7 ± 0.9	n.d.	6.5 ± 1.2
U-87MG	1.9 ± 0.2	7.8 ± 1.0	1.7 ± 0.4
U-138MG	2.4 ± 0.5	9.8 ± 2.4	1.6 ± 0.5
U-343MG	2.7 ± 1.3	n.d.	0.8± 0.7

^1^ Cell cultures in 96-wells were infected by 150 µL of LV-containing cell culture supernatants. The GFP+ rate in % and the standard deviation were calculated by counting the GFP+ and DAPI+ cells in each of the four 96-wells. Figure A2 shows an example of a section of a LV transduced and DAPI stained culture. n.d., not done.

## Data Availability

The authors confirm that the data supporting the findings of this study are all available within the figures and tables of the article.

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
