# Peer review of "Transduction Efficiency of Zika Virus E Protein Pseudotyped HIV-1gfp and Its Oncolytic Activity Tested in Primary Glioblastoma Cell Cultures"

_cancers, 2024, doi:10.3390/cancers16040814_

Round 1

Reviewer 1 Report

Comments and Suggestions for Authors

Overall, this is a very novel approach to the targeting of oncolytic viruses to glioblastoma cells by psueotyping HIV-1gfp with Zika virus E protein. The results are well-presented. My main concern is that it seems likely that this virus would infect neural stem cells and similar "normal" undifferentiated cell types in the brain, leading to potential "off-target" effects. (However, this is not massive concern given the dismal 5-year survival for GBM.)  The experiments here on normal cells is not suffient to exclude potential effects on the heterogeneous cell populations in the brain (e.g. https://pubmed.ncbi.nlm.nih.gov/28736217/; https://actaneurocomms.biomedcentral.com/articles/10.1186/s40478-017-0450-8; https://www.ncbi.nlm.nih.gov/pmc/articles/PMC5097023/). I would request that the authors add a bit more to the discussion regarding these potential caveats though. I would also call attention to a directly pertinent recent paper on Zika as an oncolytic agent in GBM: (https://pubmed.ncbi.nlm.nih.gov/37830597/)

Author Response

Reviewer 1 Overall, this is a very novel approach to the targeting of oncolytic viruses to glioblastoma cells by pseudotyping HIV-1gfp with Zika virus E protein. The results are well-presented. My main concern is that it seems likely that this virus would infect neural stem cells and similar "normal" undifferentiated cell types in the brain, leading to potential "off-target" effects. (However, this is not massive concern given the dismal 5-year survival for GBM.)  The experiments here on normal cells is not sufficient to exclude potential effects on the heterogeneous cell populations in the brain

(e.g. https://pubmed.ncbi.nlm.nih.gov/28736217/; https://actaneurocomms.biomedcentral.com/articles/10.1186/s40478-017-0450-8; https://www.ncbi.nlm.nih.gov/pmc/articles/PMC5097023/).

I would request that the authors add a bit more to the discussion regarding these potential caveats though. I would also call attention to a directly pertinent recent paper on Zika as an oncolytic agent in GBM: (https://pubmed.ncbi.nlm.nih.gov/37830597/)

Response:

Thank you for your comments on the off-target issue. We have included a section on this topic in the discussion section at the end of paragraph 4.1.

It is of course another important goal to investigate LVs and their efficiency for normal brain cells to study off-target effects in more detail. To this end, we plan to conduct further experiments in the future. However, this will require additional ethical approvals, which we will apply for shortly. Please see lines 455-477

Reviewer 2 Report

Comments and Suggestions for Authors

In this study, Formanski and colleagues evaluated transduction efficiency, selectivity, and oncolytic activity of Zika virus E protein pseudotyped HIV-1gfp for primary glioblastoma cells in vitro. The authors demonstrated that ZIKV/HIVgfp has 4-6 folds higher transduction efficiency for GBM cells compared to the commonly used VSV/HIVgf.  Importantly, Axl/Gas6 and integrin avb5, the potential ZIKV binding receptors, can be found regularly on GBM tumor cells. ZIKV/HIVgfp demonstrated high selectivity for tumor cells with stemness characteristics like expression of Nanog, Nestin, Oct4, and Sox2 compared to normal cells. This study significantly advanced development of the new oncolytic tools against GBM, the most aggressive form of brain cancer. The work has a strong potential for clinical implications. I recommend the manuscript for publication after minor revision. Bellow I have a couple of suggestions on technical details and some recommendations to improve manuscript:

1. It will be important to evaluate/or mention the transduction efficiency of ZIKV/HIVgfp for tumor neurospheres. Does your cell culture represent only adherent cells? Tumor neurospheres reflect tumor stemness and renewal potentials and may express different receptor types compared to adherent cells and could be less or more sensitive to transduction by ZIKV/HIVgfp.

2. For better data visualization, please include graphs with quantitative expression of Axl/Gas6 and integrin avb5 receptors on evaluated tumor cells. Did you observe any correlation between Nanog, Nestin, Oct4, Sox2 expression and Axl/Gas6, integrin avb5 receptor expression?

3. Evaluation of transduction efficiency at the hypoxic condition (to mimic GBM microenvironment) versus normal conditions for primary GBM cells or established cell lines in vitro will be valuable.

4. Please include this citation: Zhu Z, Gorman MJ, McKenzie LD, Chai JN, Hubert CG, Prager BC, Fernandez E, Richner JM, Zhang R, Shan C, Tycksen E, Wang X, Shi PY, Diamond MS, Rich JN, Chheda MG. Zika virus has oncolytic activity against glioblastoma stem cells. J Exp Med. 2017 Oct 2;214(10):2843-2857. doi: 10.1084/jem.20171093. Epub 2017 Sep 5. Erratum in: J Exp Med. 2017 Sep 15;: PMID: 28874392; PMCID: PMC5626408.

Technical questions:

1.         Please include the time point of ZIKV/HIVgfp collection after COS-1 cell transfection (lines 177-178). Is it 24-48 hours?

2.         Please explain the purpose of anti-PD1 antibody usage during LV input testing (line 215).

3.         The LV yield in COS-1 cell culture supernatants was in the range of 1.2-0.8 ng p24/ml for ZIKV/HIVgfp and 0.8-1.0 ng p24/ml for G-HIVgfp pseudotypes.  Line 196-197

Please clarify if the virus titer is calculated. Was the p24 value used for the normalization of virus values during cell transduction? Probably, it will be possible to calculate virus titer based on G-HIVgfp titer values in combination with p24 value.

4.         Cell viability assay (like using PrestoBlue cell viability reagent) in addition to the GFP imaging to evaluate cell survival in control versus treatment will be a helpful alternative.

Author Response

Reviewer 2 In this study, Formanski and colleagues evaluated transduction efficiency, selectivity, and oncolytic activity of Zika virus E protein pseudotyped HIV-1gfp for primary glioblastoma cells in vitro. The authors demonstrated that ZIKV/HIVgfp has 4-6 folds higher transduction efficiency for GBM cells compared to the commonly used VSV/HIVgf.  Importantly, Axl/Gas6 and integrin avb5, the potential ZIKV binding receptors, can be found regularly on GBM tumor cells. ZIKV/HIVgfp demonstrated high selectivity for tumor cells with stemness characteristics like expression of Nanog, Nestin, Oct4, and Sox2 compared to normal cells. This study significantly advanced development of the new oncolytic tools   implications. I recommend the manuscript for publication after minor revision. Bellow I have a couple of suggestions on technical details and some recommendations to improve manuscript:

  1. It will be important to evaluate/or mention the transduction efficiency of ZIKV/HIVgfp for tumor neurospheres. Does your cell culture represent only adherent cells? Tumor neurospheres reflect tumor stemness and renewal potentials and may express different receptor types compared to adherent cells and could be less or more sensitive to transduction by ZIKV/HIVgfp.

Response: Yes, our cell culture model is based on adherent cells. A 3D cell culture model. It is not sure if such a model is useful for LVs as they are not replicating viruses and LVs may not get to the cells inside the spheres. However, the idea to test LVs in 3D-modells is interesting and it will certainly be important to consider this cell culture system in future studies. We are currently working with a 2-D model with adherent cells, which has the advantage of being able to study the entry of LVs well, but also has the corresponding disadvantages compared to a 3-D model. We have not explicitly mentioned the 2-D model (lines 440-444) but have justified the importance of the adherent cell cultures system for our analyses (lines 446-454).

  1. For better data visualization, please include graphs with quantitative expression of Axl/Gas6 and integrin avb5 receptors on evaluated tumor cells. Did you observe any correlation between Nanog, Nestin, Oct4, Sox2 expression and Axl/Gas6, integrin avb5 receptor expression?

Response: Unfortunately, we are not able to include such a quantitative analysis in the manuscript. However, we can say that Axl/Gas6 is predominant in all GBM cells that we have cultured. This is not the case for integrin avb5. Integrin-positive cells are significantly less present in our GBM cultures. At the moment we do not have access to FACS sorting, which would be necessary for the analysis. In the next experiments we will definitely try to investigate such a correlation. Since the GBM cells lose their morphology very quickly, we do not consider it very likely that GFP is a suitable marker for FACS correlation studies. We are currently working on an LV with an E-crimson fluorescence marker which, according to the literature, should be a more gentle marker for brain cells.

In regard to ZIKV receptors, our experiments do not allow us to decide which of the two ZIKV receptors seems to be crucial for ZIKV entry. In mice, Axl is not crucial, since infection proceeds normally in Axl -/- mice, whereas in cell lines, blocking Axl prevents ZIKV infection. Integrin is also an important factor and ZIKV entry can be blocked by integrin antibodies. But this is also preferentially shown in cell lines.

We have added a comment on our Axl-integrin experience in the manuscript discussion (lines 424-426).

  1. Evaluation of transduction efficiency at the hypoxic condition (to mimic GBM microenvironment) versus normal conditions for primary GBM cells or established cell lines in vitro will be valuable.

Response: Thank you very much for this comment and the idea of investigating this type of effect. We will consider this in our next studies.

  1. Please include this citation: Zhu Z, Gorman MJ, McKenzie LD, Chai JN, Hubert CG, Prager BC, Fernandez E, Richner JM, Zhang R, Shan C, Tycksen E, Wang X, Shi PY, Diamond MS, Rich JN, Chheda MG. Zika virus has oncolytic activity against glioblastoma stem cells. J Exp Med. 2017 Oct 2;214(10):2843-2857. doi: 10.1084/jem.20171093. Epub 2017 Sep 5. Erratum in: J Exp Med. 2017 Sep 15;: PMID: 28874392; PMCID: PMC5626408.

 Response: added to the introduction, line 88

Technical questions:

  1. Please include the time point of ZIKV/HIVgfp collection after COS-1 cell transfection (lines 177-178). Is it 24-48 hours?

 Response: LV containing supernatants were harvested 72 h post transfection, added to line 177

  1. Please explain the purpose of anti-PD1 antibody usage during LV input testing (line 215).

 Response: The anti PD-1 antibody was available in high amounts from leftovers in infusion bags that are going into trash. We have used this antibody just as a mAb for quenching any kind of unspecific antibody effect in the test wells.

We have added a comment on that in the Mat & Met section 2.6. Line 218

  1. The LV yield in COS-1 cell culture supernatants was in the range of 1.2-0.8 ng p24/ml for ZIKV/HIVgfp and 0.8-1.0 ng p24/ml for G-HIVgfp pseudotypes.  Line 196-197

Please clarify if the virus titer is calculated. Was the p24 value used for the normalization of virus values during cell transduction? Probably, it will be possible to calculate virus titer based on G-HIVgfp titer values in combination with p24 value.

Response: At the moment the p24 value is the only valuable test that give us some information on the amount of LV particles in the COS-1 cell culture supernatants. We check our LV supernatants for p24 since this gives us a good hint if the transfection has been successfully in regard to LV production. Since ZIKV/HIV and VSV/HIV are very different in assembly/budding and also it is not clear if the have a similar tropism, testing for p24 seems to be the most suitable method (at the moment) for obtaining comparative results.

Please see line 199-200: Equal amounts of p24 equivalents were used in comparative transduction experiments.

  1. Cell viability assay (like using PrestoBlue cell viability reagent) in addition to the GFP imaging to evaluate cell survival in control versus treatment will be a helpful alternative.

Response: Thank you very much for this technical tip.